# A New Pharmacological Vitreolysis through the Supplement of Mixed Fruit Enzymes for Patients with Ocular Floaters or Vitreous Hemorrhage-Induced Floaters

**DOI:** 10.3390/jcm11226710

**Published:** 2022-11-13

**Authors:** Jui-Wen Ma, Jen-Lin Hung, Masaru Takeuchi, Po-Chuen Shieh, Chi-Ting Horng

**Affiliations:** 1Department of Pharmacy, Tajen University, Pingtung 907, Taiwan; 2Unique Biotechnology Co., Ltd., Kaohsiung 800, Taiwan; 3Graduate Institute of Health Care, Meiho University, Pingtung 912, Taiwan; 4Department of Ophthalmology, National Defense Medical College, Saitama 359-8513, Japan; 5Department of Ophthalmology, Fooying Unversity Hospital, Pingtung 928, Taiwan

**Keywords:** ocular floater, vitreous hemorrhage, bromelain, papain, ficin, pharmacologic vitreolysis

## Abstract

Purpose: Ocular floaters caused by vitreous degeneration or blood clots may interfere with various visual functions. Our study investigated the pharmacologic effects of oral supplementation of mixed fruit enzymes (MFEs) for treating spontaneous symptomatic vitreous opacities (SVOs) and those secondary to vitreous hemorrhage (VH). Methods: 224 patients with monocular symptomatic vitreous opacities (SVOs) were recruited between September and December 2017 and received oral supplementation of MFEs (190 mg bromelain, 95 mg papain, and 95 mg ficin) for 3 months in a double-blind clinical trial. Participants were divided according to the etiology of the SVOs, spontaneous (experiment 1) versus VH (experiment 2), and then randomly assigned into four treatments groups: one group received oral vitamin C, as a placebo; and the other 3 groups received 1 capsule per day (low dose), 2 capsules per day (middle dose), or 3 capsules per day (high dose) of MFEs. The number of SVOs was determined at baseline and then 1, 2, and 3 months after initiating treatment. Further, in cases secondary to VH, the changes in corrected distance visual acuity (CDVA) were assessed after 3 months. Second, we compared the free radical scavenging capabilities of each substance: vitamin C, bromelain, papain, ficin, and MFEs (combination of bromelain, papain, and ficin) by DDPH assay. Finally, SVOs-related symptoms and satisfaction with the treatments were evaluated at the last follow-up visit Results: In experiment 1, the disappearance rate of SVOs was 55%, 62.5%, and 70% after taking 1, 2, and 3 capsules daily, respectively (total *p* < 0.001), in a dose-dependent manner. In experiment 2, the disappearance rate of VH-induced SVOs was 18%, 25%, and 56% (*p* < 0.001) after 1, 2, and 3 capsules of the supplement daily, respectively. Additionally, the patients’ vision elevated from 0.63LogMAR to 0.19LogMAR (*p* = 0.008). Conclusions: A pharmacological approach using a high dose of oral supplementation with MFEs (bromelain, papain, and ficin) was effective in reducing vitreous opacities, even after intraocular hemorrhage. Furthermore, pharmacologic vitreolysis with MFEs supplementation showed high patient satisfaction, and also improved CDVA in patients with vitreous hemorrhage-induced floaters

## 1. Introduction

Human vitreous gel is composed of collagen fibers and a highly hydrated extracellular cellular matrix (ECM), including hyaluronic acid (HA) and chondroitin sulfate proteoglycans (CS), that maintains biodegradable and biocompatible activities. Most of the water in the vitreous is bound in the widely-spaced supporting framework of collagen fibers and HA. In the beginning, HA and CS take the vitreous collagen fibrils apart. With aging, pathological processes and oxidative stress occur, leading to HA depolymerization, water loss, vitreous liquefaction, and disorganization of the collagen fibrils. This phenomenon results in the formation of larger fibrils, which float in lacunas of liquefied vitreous. Moreover, the disordered collagen fibrils and HA mixes and generates vitreous opacities. These opacities in the vitreous are projected onto the retina and are interpreted by the brain as moving objects [1].

Collagens primarily presenting in the vitreous cavity are types II, V, XI, VI, and IX. Type VII collagen is a major component of the anchoring fibrils between tissues, and type VI collagen forms a filamentous network in most ECM [2]. Due to various conditions, the vitreous structure changes and posterior vitreous detachment (PVD) occurs. After the possible tractional force from PVD affects the junction of the vitreous-retinal interface, vitreous hemorrhage (VH) may happen, and various types of ocular floaters may appear.

Most floaters are small spots, including lines, circles, dots, flies, and cobwebs. They become apparent in bright space and blue skies, with monochrome backgrounds and few objects. Initially, the symptoms can be very bothersome, but the brain eventually adapts to them, and the patient becomes less aware of their existence, but this process can take from 1 week to several years, and in some cases the intense symptoms never diminish. Some vitreous debris would disappear spontaneously; however, much vitreous opacity persists under adaptation [3]. Ocular floaters, especially within the visual axis, may decrease visual acuity, contrast sensitivity, and the quality-of-life (QoL) with vision-dependent tasks [4,5]. These symptoms are not the only outcome of vitreous condensation, but it coexists with other secondary alterations in the retina. Bond-Taylor et al. found that when subjects had an acute onset of floaters, the prevalence of retinal tears or retinal breaks was around 14%, HV was 22.7%, and retinal detachment (RD) prevalence was up to 13% [6]. Furthermore, severe ocular floaters could be associated with various psychological distress, such as unhappy mood, mental stress, depression, and anxiety [7]. In the past, ocular floaters usually occurred among people over 40 years of age, but it seems that they are now occurring in younger people. It has been hypothesized that this is related to the blue light that is widely emitted by electronic displays, such as iPads, smartphones, and liquid crystal displays (LCDs). Chen et al. suggested that blue light could accelerate vitreous degeneration, resulting in increased vitreous opacity and increased ocular floaters, potentially affecting up to 39.7% of users [8]. Webb et al. studied 603 participants and found that 76% of young and middle-aged individuals (29.5 ± 10.7 years) reported seeing floaters. Moreover, 33% of the participants reported that ocular floaters caused noticeable impairment in visual acuity and various types of scotoma [9]. Over two-thirds of the patients with floaters had moderate to extreme difficulty reading small print and even night-driving [10].

In clinics, PVD, which implies that the vitreous is dragging away from the retina, is the most common cause of acute-onset floaters. Nian et al. demonstrated that ocular floaters were attributed to acute PVD in 83% of the eyes [11]; thus, they concluded the close relationship between ocular floaters and PVD. During vitreous separation, the mechanism of VH from PVD enhances the tractional force on the weakest point of the vitreous-retinal interface and the peripheral retinal lesions (i.e., lattice degeneration), tearing the retina or blood vessels nearby [12]. The prevalence of ocular floaters varied from 27% to 63%, with most patients not perceiving painless symptoms. The primary etiologies of floaters are age, environmental factors (i.e., UV light, blue light), high myopia, and oxidative stress [13]. Additionally, VH-induced ocular floaters are produced from diabetic retinopathy, hypertensive retinopathy, and ocular trauma. In clinics, excessive ocular floaters are too small to be observed by eye doctors. For easy analysis, “symptomatic vitreous opacity (SVO)” is proposed when ocular floaters are severe enough to result in various symptoms for three months, causing enough visual disturbance [14,15]. Tiny floaters do not impact the victims’ vision but become bothersome when near the visual axis. Some people may suffer from intermittent blurred vision, glare, and haze, leading to impairments in vision-related activities and tasks. At first, as floaters appear, human adaptation occurs in the first few weeks and the victims learn to “live with them”. Once an individual starts seeing floaters, they can be difficult to ignore, especially when stressed or fatigued. Larger SVOs decreased visual acuity, visual field sensitivity, and stereo-acuity [16,17]. Clinics found that SVOs may be associated with floaters, the overgrowth of the extra-cellular matrix (ECM), and intraocular blood clots. Further, 10% of patients with SVOs have retinal tears, 16% of SVOs showed retinal breaks, and 6.2% of SVOs resulted in RD. Further, patients with more than 10 SVOs or a cloud in front of their eyes had a high risk of developing retinal tears.

SVO is the main symptom of VH. Detailed analyses estimated an incidence of 4.8 VH cases per 10,000 person-years in Asia (Taiwan) [18]. Furthermore, it revealed a yearly incidence of 7 cases per 100,000 inhabitants in Europe. Further, VH-induced larger SVOs can result in visual field loss and poor vision. However, smaller and thinner VH only leads to minor scotoma and color deficiency. The most common causes of VH were PVD, and its associated complications include proliferative diabetic retinopathy (PDR), hypertension, and ocular trauma [19]. Moreover, PDR is the primary etiology of VH. For instance, DR accounts for 32–54% of VH in the United States. Additionally, the most common cause of VH in France is PDR (39.2%) [20]. Afterward, blood entering the vitreous results in rapid clot formation and may be followed by a slow clearance rate of 1% per day. The symptoms of patients with mild VH would improve within 1 to 2 months. The moderate VH disappeared at least six months later. Nevertheless, under spontaneous absorption, the possibility of retinal tears, retinal breaks, or RD may happen. Thus, aggressive treatment may be considered if VH persists. However, patients must be willing to accept a 7% risk of blindness to eliminate SVOs [21]. Therefore, providing a safer and more effective therapy is essential.

Several methods were proposed for treating SVOs. First, observation is the common strategy for handling SVOs [22]. Nevertheless, the disappearance rate of SVOs is beyond physicians’ prediction. Second, Nd-YAG lasers are another choice; this mechanism is to vaporize vitreous opacities. However, this procedure generally places patients under topical anesthesia during treatments. Unfortunately, it has many potential risks of elevated IOP, posterior capsule defect, retinal hemorrhage, and RD [23,24,25]. Third, as for VH-induced SVOs, argon laser is the best instrument for patients with VH by decreasing bleeding [26]. However, the retinal scars from the laser may result in lower contrast sensitivity, scotoma, and even vision loss [27]. If VH persisted for 3 to 6 months, PPVT may be adopted together. During this procedure, vitreous opacities, floating debris, and blood clots may be removed. However, the dangerous risks from general anesthesia should be noted. Meanwhile, cataract, increased IOP, endophthalmitis, and RD were also found. Now, pharmacologic vitreolysis may use diverse enzymatic and non-enzymatic agents to facilitate the induction of liquefying the gel and preventing PVD. For example, the vitreoretinal interface could be separated by Bevacizumab (Avastin^®^). Thus, treating VH by pharmacologic vitreous liquefaction also becomes a trend of the future.

Recently, we developed a new method involving pharmacologic vitreolysis by combining bromelain, papain, and ficin, with which we successfully treated patients with SVOs [28]. In this study, we further focused on the effectiveness of MFE capsules for patients with both primary and VH-induced SVOs. Furthermore, we evaluated the antioxidant ability of individual fruit enzymes and MFEs. Moreover, we assessed with a questionnaire patients’ impression of ocular floaters, influence level of daily activities, and satisfaction with MFEs

## 2. Methods

### 2.1. Design

We designed the double trial for human study and anti-oxidant assay (1,1diphenyl-2 picrylhydrazyl (DPPH)) test for evaluation of vitamin C, bromelain, papain, ficin, and MFEs by a 1,1diphenyl-2 picrylhydrazyl (DPPH) test. Additionally, we also assessed the SVO-induced subjective sensation and level of interference of various daily activities. Lastly, we investigated patients’ appraisal and personal satisfaction after three months of treatment. All experiments were performed between September 2017 and December 2017. We tried to analyze the efficacy of MFEs in reducing SVOs and VH-induced SVOs. If SVOs significantly disappeared, this therapy should be considered a “success”. The disappearance rate of SVOs indicated the effectiveness of MFEs treatment.

### 2.2. Subjects

All 224 participants, aged between 30 and 60, were suffering from ocular floaters, and visited Kaohsiung Armed Forced General Hospital (Kaohsiung, Taiwan) for medical help. Informed consent was obtained from each subject before participation. All experimental protocols were conducted following the Declaration of Helsinki, and ethical approval for this study was obtained from the Institutional Review Board of Kaohsiung Armed Forces General Hospital (KAFGH-106-003). In our research, SVOs and VH-induced SVOs were included. However, posterior uveitis, posterior lymphoma, sickle cell retinopathy, high myopia (>−8.0 diopters(D)), asteroid hyalosis, status post-cataract operation, RD, and endophthalmitis were excluded because of complicated vitreous-retinal pathophysiology [29].

### 2.3. Materials

The chemical substances in our method employing pharmacologic vitreolysis are derived from three natural enzymes: bromelain from pineapple, papain from papaya, and ficin obtained from latex from the trunk of a tropical fig tree. The suppliers of the purified substances were: for bromelain (powder), Xian Sgonek Biological Technology Co., Ltd. (China, Xi’an); for papain (powder), Shaani Hongda Phytochemistry Co., Ltd. (China, Xi’an); and for ficin (extract), HK. Gotopharm Co., Ltd. (China, Shenzhen). According to our previous study, each capsule was prepared to contain 190 mg bromelain, 95 mg papain, and 95 mg ficin [28,30]. All capsules were manufactured with hydroxyl propyl methylcellulose and pectin. The enteric release function from the components of this capsule could prevent gastric acid–induced fruit enzyme damage, and easily enhance absorption from the small intestine. Afterward, we evaluated whether the pharmacologic functions of MFEs benefit SVOs or not.

### 2.4. Procedure

According to the etiologies of ocular floaters, the human studies were divided into two parts. In experiment 1, 160 subjects with SVOs from PVD and other vitreous opacities were enrolled and randomly separated into four groups (each group consisted of 40 volunteers) by rolling the dice. In experiment 2, 64 patients with VH-induced SVOs were recruited, and the etiologies were either DR, trauma, or hypertensive retinopathy. All 64 participants were divided into 4 groups (each group consisted of 16 subjects). Additionally, the blood sugar and blood pressure of the patients with DM or hypertension was well controlled by medical treatment throughout the study. Therefore, there were four groups in experiments 1 and 2, including Group 1 (oral vitamin C: 10 mg/day, placebo), Group 2 (1 capsule/day) (low dose group; LDG), Group 3 (2 capsules/day) (middle dose group; MDG), and Group 4 (3 capsules/day) (high dose group; HDG). All participants took the designed number of capsules for three months. Subsequently, we checked the numbers of SVO of each subject in the first, second, and third months. Furthermore, we also compared the change in corrected distance visual acuity (CDVA) after 3-month MFEs intake in experiment 2 [31].

In real life, many individuals perceive a vitreous floater, yet ophthalmologists do not find any abnormality in the vitreous cavity. Thud, the numbers of SVOs could not be easily objectively identified. To avoid bias, during experiments 1 and 2, we selected the participants with subjective floaters that were simultaneously detected with ocular diagnostic devices. Therefore, the definition of patients’ “SVOs” describing vitreous floaters to be large enough to cause patients’ perception was employed. The vitreous opacity was evaluated by non-mydriatic retinal photography (Kowa, E-vision Co., Ltd., Japan, Osaka), B-mode ultrasonography (Ultra Scan-B, Alcon Co., Ltd., TX, USA), and optic coherence tomography (OCT, Clinico Co., Ltd., Japan, Tokyo) The images in the vitreous of volunteers must be detected by at least one of the above three instruments. First, the numbers of floaters were all recorded at the baseline. Then, we checked the numbers of SVOs monthly. If the number of floaters significantly decreased after treatments, our method would be considered a “success”. Furthermore, BCVAs were translated to the logarithm of the minimum angle of resolution-converted visual acuities (Log MAR), which may be easily standardized for analysis. The initial and final BCVAs were compared after therapies. If the number decreased, it would indicate that the VH-related SVOs diminished (experiment 3).

### 2.5. Statistical Analysis

We recorded the changes in the numbers of SVOs before and after the MFE supplement for 3 months. The data of SVOs of every volunteer were obtained on a monthly basis to record the disappearance rate of SVOs and verify the effectiveness of MFEs.

The values were expressed as mean ± SD. All the results could be analyzed with SPSS version SAS 25 (SAS Inst., Cary, NC, USA). We compared the outcome between taking MFE capsules during the initial month, each month, and three months later by Scheffe tests. Each month, we also compared the results of LPG, MPG, and HPG with the placebo group by the Williams’ test; when *p* < 0.05, the differences were considered statistically significant.

### 2.6. 1,1 Diphenyl-2 Picrylhydrazyl (DPPH) Test

Recently, the DPPH test has been primarily used to monitor chemical reactions involving radicals. Most notably, it can be used to evaluate a common antioxidant assay. DPPH belongs to a stable radical and a trap “scavenger” for other radicals. When it is soluble in methanol or ethanol, the fluid shows a blue-purple color. The substances should be added and mixed with DDPH. The solution initially changes into colorless or pale yellow when neutralized. According to the principle, the less yellow color of the mixed solution reveals good antioxidant abilities. Additionally, strong optical absorption bands are centered at 517–520 nm, in which a spectrometer must be used to identify the capacity. At the beginning, we utilized a DPPH test to realize the antioxidant ability of bromelain, papain ficin, MFEs, and vitamin C. All procedures were performed at the Institute of Biological Chemistry, Chung Hsing University (Taichung, Taiwan) by the same researcher (Ma Ph.D.). First, 50 μL of test materials (i.e., bromelain power) in methanol (the final concentrations were 1, 5, 10, 20 and 40 g/mL, respectively) were mixed with 450 μL of 50 mM. Tris-HCL buffer (PH:7.4) and 1500 μL mM of 0.1 m DPPH-ethanol solution. After 30 min of incubation, the reduction rate of DPPH was measured by reading the absorbance at 517 nm under a spectrometer. Furthermore, the inhibition ratio was calculated using the following equation: (%, inhibition) = (absorbance of controlled absorbance of test control) × 100 [32]. Subsequently, a diagram was drawn: the horizontal direction presented the concentration (mg/mL), and the results in the vertical direction were the scavenging effect (%). Additionally, papain, ficin, MFEs, and vitamin C were arranged to obtain antioxidant values by DDPH test, in which all procedures were the same as in the test method for bromelain.

### 2.7. Questionnaire on Satisfaction and Subjective Sensation

After the three-month treatment, we designed six questions associated with the subjective feelings towards ocular floaters. In particular, we surveyed the effectiveness of drugs and patients’ satisfaction.

## 3. Results

A total of 224 patients (124 right eyes and 100 left eyes) were recruited, including 126 male and 98 female subjects. Their mean age was 43.4 ± 16.4 years old. Additionally, there was no apparent retinal damage, optic nerve disorders, corneal defects, cataract formation, or ocular inflammation found during the three months. Noteworthily, no patients felt unwell after the treatments.

### 3.1. Human Study

In experiment 1, with 160 subjects with SVOs, we compared the results of MFEs in treating SVOs after three months. The rates of disappearance in SVOs were 55%, 62.5%, and 70% in the 1-, 2-, and 3- capsule groups after three months (*p* < 0.05), respectively (Table 1). Therefore, we demonstrated that SVOs should be broken down and absorbed by MFEs. Moreover, the outcomes showed a dose-dependent manner. In other words, the higher amounts of fruit enzyme in the capsules, the stronger the hydrolytic and proteolytic ability for dissolving SVOs. If the patients took at least one capsule of MFEs, SVOs might significantly disappear. Further, the subjects in the placebo were given vitamin C for three months; however, SVOs apparently persisted (*p* > 0.05). For example, a 55-year-old female patient complained about SVOs from a B-scan (Figure 1A). After a three-month treatment (3 capsules/day), the vitreous floater remarkably disappeared (Figure 1B).

Initially, we compared the results at the beginning and during the entire 3 months by conducting the Scheffe test in the four groups. Statistical significance was considered at *p* < 0.05 (represented by *). Then, we compared the prognosis in the 3rd month (Group 2, Group 3, and Group 4) with the placebo group by the Williams’ test. A *p*-value of less than 0.05 indicates a significant difference (represented by #).

In experiment 2, 64 patients with VH-induced ocular floaters suffered from dark-black and even brown-red colors of cloud-like shapes blocking the visual axis. Similarly, we compared the results after three months, in which only HDG revealed a significant disappearance rate (66%), as compared with the baseline (*p* < 0.05) (Table 2). Furthermore, we analyzed the change in BCVA at the baseline during a three-month period. We found that only HDG could enhance the improvement in patients’ vision. Patients with VH after HDG (3 capsules daily) intake increased to 0.19 LogMAR compared with the baseline (0.63 log MAR) (*p* < 0.05) (Table 3). For example, one patient aged 58 with DM for several years presented an obvious VH. He complained about a massive ocular floater in front of his visual axis. The patient took three capsules daily, and after three months, the VH significantly disappeared. The intraocular blood clots were cleared and gradually absorbed, allowing the light to pass through the vitreous cavity again. Finally, the patient’s BCVA elevated remarkably from 0.52 LogMAR to 0.18 log MAR (Figure 2A,B).

We checked the outcomes at the baseline of the study and during the whole 3 months by conducting the Scheffe test in the four groups. Statistical significance was considered at *p* < 0.05 (represented by *). We also compared the results in the 3rd month (Group 2, Group 3, and Group 4) with the placebo group by the Williams’ test. A *p*-value of less than 0.05 indicates a significant statistical difference (represented by #).

Initially, we compared the results at the beginning and during the whole 3 months by conducting the Scheffe test in the four groups. Statistical significance was considered at *p* < 0.05 (represented by *). Then, we compared the prognosis in the 3rd month (Group 2, Group 3, and Group 4) with the placebo group by the Williams’ test. A *p*-value of less than 0.05 indicates a significant difference (represented by #).

### 3.2. The Evaluation of Antioxidant Abilities (DDPH Test)

The free radical scavenging activity of bromelain, papain, ficin, MFEs, and vitamin C (placebo) was assessed by DDPH assay. Figure 3 shows, at various concentrations (1, 5, 10, 20, 40), the antioxidant effects (free radicals scavenging effects) MFEs > bromelain > vitamin C > ficin > papain. Surprisingly, the antioxidant capacity of MFEs showed outstanding ability levels. For instance, when the concentration of MFEs was about 40 mg/mL, the free radicals scavenging rate reached 80%. In other words, taking MFEs could apparently dissolve and disappear SVOs.

### 3.3. Questionnaire on Satisfaction and Subjective Sensation

During the visit, 80% of the subjects felt a level of difficulty of very severe due to SVOs without treatment, and 78% of patients always and often had difficulty performing various daily activities due to SVOs. Additionally, 90% of the subjects with SVOs were affected while reading; 81% became dangerous when driving; 60% were often interrupted during work by SVOs. Further, 82% of the patients felt “brighter” after the second month, 92% felt “bright” after three months; 90% felt much better after the three-month therapy. Finally, the patients’ satisfaction with MFEs therapy after three months was approximately 92% (Figure 4).

## 4. Discussion

The vitreous body is a type of transparent ECM mainly composed of H_2_O (98%), glycosaminoglycans (i.e., HA, chondroitin sulfate, heparin sulfate), opticin, versican, and collagen fibers. The gel state of the vitreous is maintained by a network of long, thin collagen fibers that are ~15 nm in diameter, organizing types II (the most abundant). Furthermore, collagen fiber types I, III, IV, VIII, XII, XIV, and XVI are found to endow the shape, strength, flexibility; and resist the tractional forces and maintain the gel structure. There are almost 27 types of collagen molecules identified to date. HA levels significantly decrease with aging. Meanwhile, free radicals generated by oxidative stress could trigger the dissociation of swollen fibers and HA, ultimately resulting in vitreous liquefaction [33]. Afterwards, vitreous opacities developed including reduced viscoelasticity, molecular alterations, liquid-filled space, optically dense condensations within the matrix and the formation of PVD [34,35]. Indeed, acute PVD may develop into retinal tears, retinal hemorrhage, VH, and RD. The disruption in the HA-collagen complex could cause the collagen fibrils to aggregate into bundles, which become large enough to be visible and identified as SVOs. Thus, vitreous liquefaction and the collapse of the matrix could be followed by PVD, leading to the formation of SVOs that vitreous shrink and gradually result in the clumping of vitreous balls impressed on the retina [36,37].

The occurrence of SVOs has two main causes: ocular hemorrhage and the degenerative rearrangement of collagen fibrils. Schulz-Key et al. proposed that common SVOs includes vitreous liquefaction, flecks of protein, vitreous debris, and inflammatory balls [38]. Intraocular hemorrhage consists of damaged collagen fibrils, the dense matrix of the posterior vitreous cortex, and blood clots due to various retinopathy and ocular trauma, blocking the visual axis and decreasing human vision. Thus, the relationship among ocular floaters, PVD, and VH is close. Moreover, the etiologies of VH-related SVOs include retinal tears, retinal hemorrhage, PDR, ocular trauma, neovascularization from retinal vein occlusions, RD, and hypertension [39]. Additionally, subjects who were prescribed anti-coagulation drugs and health food supplements, including aspirin, warfarin, clopidogrel, astaxanthin, deep sea oil, and Ginkgo Biloba, were prone to bleeding tendencies. The primary cause of VH is PDR, which induces severe visual loss and even blindness. The obvious VH-induced SVOs limit visual acuity or cause various scotomas in patients. Our research found the subjects with VH-induced SVOs showed a reduction of vision. The mean BCVA decreased to 0.61Log MAR (Table 3). Thus, we recommended that individuals under higher risk of SVOs or VH formation should be followed up by ultrasonographic and surgical expertise at regular and even emergent times.

There are several methods of therapies for SVOs and VH-induced SVOs, including simple observation, lasers (Nd: YAG laser for SVOs and Argon laser for VH-induced SVOs), PPVT, and pharmacologic vitreolysis. Observation is the first and most reasonable choice for SVOs. If patients have few and smaller SVOs, eye doctors suggest waiting for their spontaneous dissolution. However, the success rate cannot be predicted, and larger SVOs often do not disappear themselves. As observed, ocular vitreous spontaneously disappeared only among 11% of patients, whereas 64% of subjects with floaters lost at least 2 lines by Snellen lines within 5 years. Based on the principle of VH-induced SVOs, most ophthalmologists suggested waiting for absorption within 3–6 months. Thus, waiting for the disappearance of VH for several months has become the commonly employed strategy. It was reported that VH did not spontaneously diminish within half a year and would develop to various degrees of fibrovascular proliferation, which is prone to retinal breaks or tractional RD and neovascular glaucoma [40].

Laser and PPVT are other options for treating ocular floaters. For example, the Nd-YAG laser is suggested for patients with common SVOs that are non-invasive and effective for treating opacities in the anterior and mid-vitreous cavity [38]. However, if SVOs are too near the retina, this laser becomes dangerous because the power may damage the retina and leads to retinal hemorrhage and maculopathy [41]. Furthermore, the use of the YAG laser has achieved highly variable success rates. However, unpredictable outcomes sometimes disappoint patients. The laser procedure may result in many serious side effects, including RD, cataract, retinal tear or hemorrhage, scotoma, and elevated IOP. On the other hand, the argon laser is another medical method for treating VH-induced SVOs. After waiting for a period of time of spontaneous absorption and after the intraocular hemorrhage mildly clears, the fundus reflex may become brighter [42]. At this time, we might find the sources of VH and use the argon laser beam to repair the leakage of retinal blood vessels that stops hemorrhage as soon as possible. However, the over-power of the argon laser may damage the retinal cells, decreasing the contrast sensitivity and visual acuity, and impairing the visual field sensitivity.

PPVT is considered the better option for completely clearing the more diffuse vitreous, especially SVOs within the visual axis, resulting in the significant disturbance of visual function. Subjects who experience persistent, larger, and severe SVOs should undergo PPVT. Poster-operative Snellen visual acuity would increase (8–44 months follow-up), and some patients expressed satisfaction with the outcome [43]. However, despite PPVT being a good method for SVOs, some complications limit the broad indication—for instance, endophthalmitis, cystoid macular edema, retinal tear, cataracts, and IOP elevation [44].

Pharmacological vitreolysis reduces or eliminates the pathogenetic role of the vitreous by drug delivery. Recently, intravitreal injection of ocriplasmin (microplasmin) is considered a new strategy for treating various SVOs only in some countries. However, it did not gain prominence because of its reported disadvantages [45]. Ocriplasmin is a recombinant and truncated form of serine protease plasmin that cleaves collagen, extra-ECM, vitreous fibrils, and various structural proteins, such as laminin and fibronectin. Recently, ocriplasmin could be artificially synthesized for clinical use that intra-vitreal injection would cut SVOs and proliferative tissues by the proteolytic and antioxidant functions due to ocriplasmin that showed dose- and time-dependent cleavage. Meanwhile, Stalmans et al. also revealed the significant resolution of SVOs was achieved at 26.5% on day 28, remarkably better compared with only 10.1% in the placebo [46]. However, the disappearance of SVOs needs an unpredictable repeated injection, which may be harmful to the patients including local pain, intragenic cataract, intraocular inflammation, increased IOP, bleeding, retinal tear, and RD. Surprisingly, ocriplasmin was also accidentally discovered in ficin extract. Thus, we believed that ficin intake could be used to dissolve and absorb SVOs [28,30]. Additionally, it was interesting to find MFEs in our designed capsule also contained the component of ficin. Therefore, we proposed that our MFEs supplementation containing ocriplasmin are safer for treating SVOs than injecting ocriplasmin.

Fruit enzymes are popular in the food industrial, agricultural, and medical fields. The proteolytic enzymes, including papain, bromelain, ficin, zingibain, and actinidin are widely used [47]. Also well known are the frequent consumed proteolytic enzymes papaya, pineapple, ginger, and kiwi. However, the concept and application of pharmacologic vitreolysis by various fruit enzymes have never been proposed. In our cross-international research between Taiwan and Japan several years ago, we developed a new method employing three MFEs in a capsule for integrating and absorbing SVOs [28].

In this study, the disappearance rates of SVOs reached almost 55%, 62.5%, and 70% after taking 1-, 2-, and 3 capsules daily, respectively, revealing a dose-dependent manner. As for VH-induced SVOs, this study revealed that taking 3 capsules of MFEs each day for 3 months could significantly reduce 56% of intraocular blood. It revealed that many proteases and collagenase were found in our MFEs, which may dissolve misarranged collagen fibrils, injured HA, blood clots, and cell debris. Therefore, we considered that the mixture of bromelain, papain, and ficin could be used for vitreolysis. First, bromelain is one of the most common fruit enzymes extracted from pineapples. Bromelain has many pharmacologic functions such as hydrolytic, antifibrinolytic, anti-inflammatory, and anti-thrombotic properties [47,48]. In clinics, bromelain is used to treat patients with osteoarthritis, sinusitis, and post-operative swelling. It may enhance hydrolysis of the damaged collagen fibers and clear blood clots in vitreous. Thus, we suggested that bromelain could decrease SVOs and VH-induced SVOs. Second, papain is a non-specific cysteine proteinase extracted from papaya that possesses the proteolytic enzymes that break down larger proteins, collagen fibrils, and abnormal ECM. Wei et al. showed that the hydrolysis of collagen by papain and bromelain by the mechanism of papain-catalyzed hydrolysis of N-acetyl-Phe-Gly 4-nitroanilide [49]. In clinics, papain is used in wound healing, aiding digestion, DM control, and bronchitis. Noteworthily, the collagenase and antioxidant activity of bromelain are nearly twice higher than papain. These reports are similar to the DDPH test in our study (Figure 4). Therefore, we proposed that papain may get rid of SVOs and clear VH by its anti-thrombotic ability [50]. Third, ficin is a fruit enzyme derived from a latex substance of the trunk of Ficus carica. It belongs to the serine proteinase family, and the mechanisms of vitreolysis are similar to ocriplasmin. Some doctors consider ficin as beneficial for maintaining blood pressure and cosmetic care for patients. Recently, it has been found that ficin could be used for hydrolytic and proteolytic degradation and solubility of elastin and collagen fibers. Additionally, ficin enhanced the enzymatic activities of bromelain and papain [51]. Therefore, MFEs have excellent hydrolytic and collagenolytic abilities to dissolve abnormal ECM, collagen fibrils, and blood clots [52].

In summary, bromelain, papain, and ficin were found to have antioxidant abilities for dealing with SVOs and VH. Therefore, we considered that various SVOs would be assimilated by MFEs. For example, bromelain has antioxidant activity including free radical scavenging and lipid peroxidation inhibition. Additionally, bromelain at 30 and 15 mg/mL were similar to glutathione (GSH). Meanwhile, polyphenols, β-carotene, and vitamin C were found in bromelain [53]. Besides, You et al. found SOD, Glutathione peroxidase (GPx), and catalase (CAT), which may eliminate ROS, were detected after papain digestion [54]. Papain inhibits peroxidation and scavenges free radicals with antioxidant titers, such as vitamin C and E [55], Further, ficin has a similar antioxidant ability characterized as peroxidase and vitamin C [56]. In a DPPH survey, we reported the free radical scavenging activities of bromelain were similar to vitamin C. Moreover, MFEs have a higher antioxidant ability that alleviates free radicals and decreases SVOs. Furthermore, matrix metalloproteinases (MMPs), which belong to the proteinases that could cut and absorb ECM, discorded collagens, and SVOs. In particular, MMP-2 and MMP-9 are primarily found in bromelain, papain, and ficin [57]. Free radical scavenging activities were found in bromelain (MMP-2), papain (MMP-2 and MMP-9), and ficin (MMP-2 and MMP-9). Simultaneously, MMPs could resect proliferative tissues and compromised collagens, which is beneficial for cleaning VH-induced SVOs under oxidase stress [58,59]. The blood-retinal barrier (BRB) consists of the inner and outer components and forms tight junctions between the retinal capillary endothelial cells and pigment epithelial cells that maintains a balanced microenvironment and prevents certain substances from entering the retina. Yang et al. found that MMP-2 and MMP-9 could reduce the tight junction proteins (i.e., claudin-5 and occlusion) and the integrity of BRB [60]. Thus, MFEs may cross BRB after taking and be absorbed by the small intestines.

The ubiquity of SVOs may sometimes affect the performance of daily activities. The findings were consistent with our results, revealing that at least 80% of the subjects felt a dofficulty level of severe due to ocular floaters, 90% of the subjects with SVOs were affected while reading, and 81% of the subjects with floaters were dangerous when driving (Figure 3). Indeed, SVOs may interfere with daily life, such as reading, distance viewing, driving, and performing near work (i.e., computer or smartphone use, writing letters). Additionally, they may affect an individual’s dynamic performance—for instance, reading sheets while playing piano, playing table tennis or badminton, and checking the peripheral vision of military pilots and taxi drivers. Additionally, SVOs may negatively affect patients’ health-related lives. It was also mentioned that the negative impact of SVOs on QoL was comparable to other diseases, including angina, mild stroke, colon cancer, ARMD, DR, and asymptomatic HIV infection [61]. Patients with SVOs who are more professionally successful and intelligent tend to notice floaters and have an increased desire to have them treated. Additionally, the patients with SVOs often get disappointed and depressed when eye doctors fail to address the health outcomes. Therefore, cleaning SVOs may improve patients’ physical and psychologic condition.

Some limitations were found in this study. First, the mechanisms of MFEs dissolved and absorbed SVO and VH need to be further investigated. Second, SVO for a few patients still persisted after three-months of MFEs treatment, as some types of collagen fibers form-ing vitreous opacity could not be absorbed and resulted in the disappearance rate of SVO below100%.

## 5. Conclusions

This study reports the results of a newly developed method that employs fruit enzymes oral supplementation to treat eye floaters. The mixed fruit enzymes, including bromelain, papain, and ficin, were effective in reducing vitreous opacities, even after intraocular hemorrhage. Furthermore, pharmacologic vitreolysis with MFEs supplementation showed high patient satisfaction, and also improved CDVA in patients with vitreous hemorrhage–induced floaters.

## Figures and Tables

**Figure 1 jcm-11-06710-f001:**
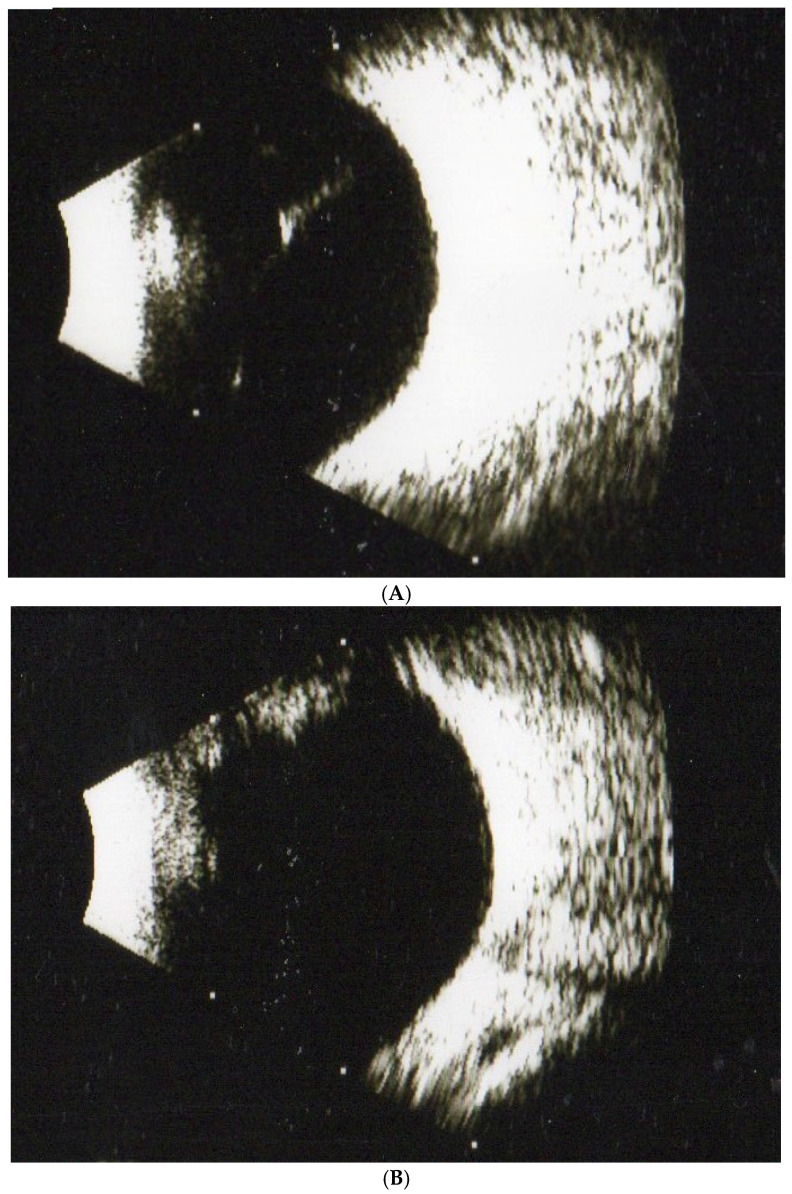
(**A**) A female patient aged 55 with one SVO was diagnosed by ultrasonography (B-scan) at the beginning of our study. (**B**) The SVO from the patient significantly disappeared after 3 months of treatment with mixed fruit enzymes as observed in the B-scan evaluation.

**Figure 2 jcm-11-06710-f002:**
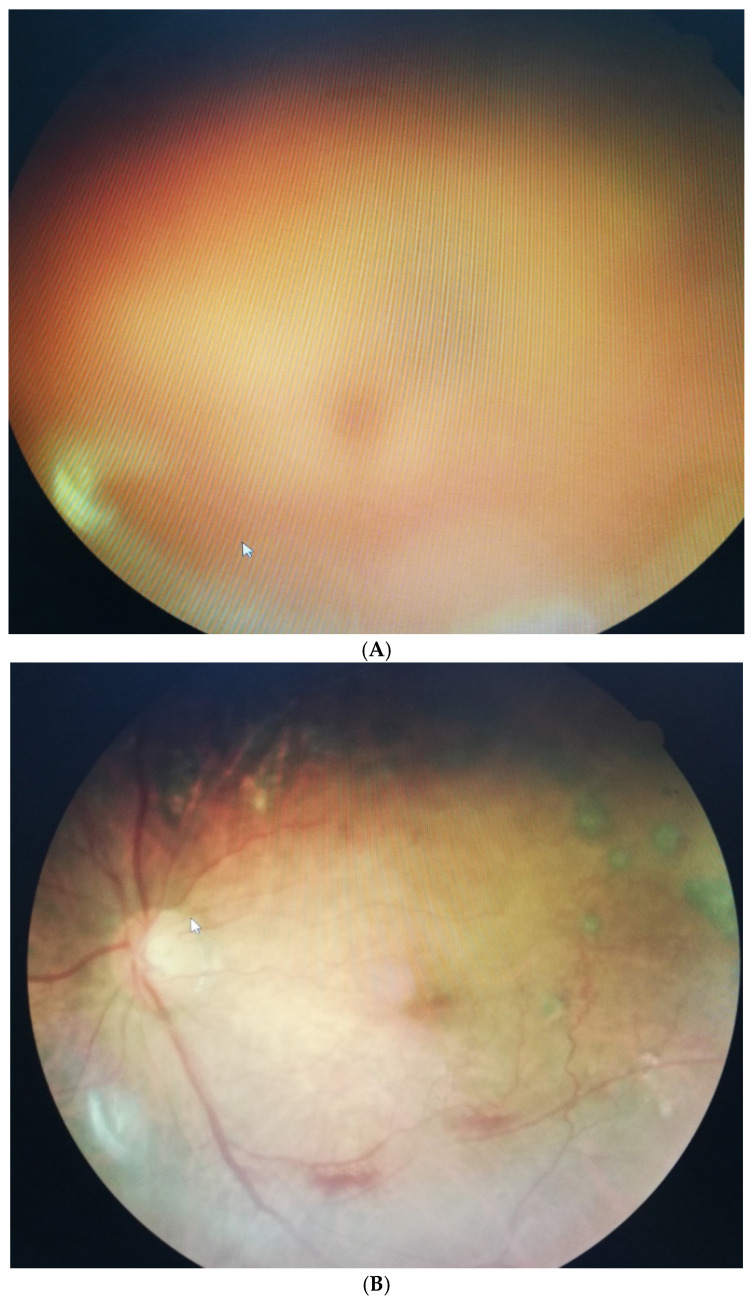
(**A**) A male DM patient aged 58 presented obvious VH-induced SVOs at the beginning of our study. The dark-red cloud overlapped the retinal fundus. (**B**) VH-induced SVOs gradually diminished after a three-month mixed fruit enzyme supplement. The patient’s vision improved from 6/60 to 30/60.

**Figure 3 jcm-11-06710-f003:**
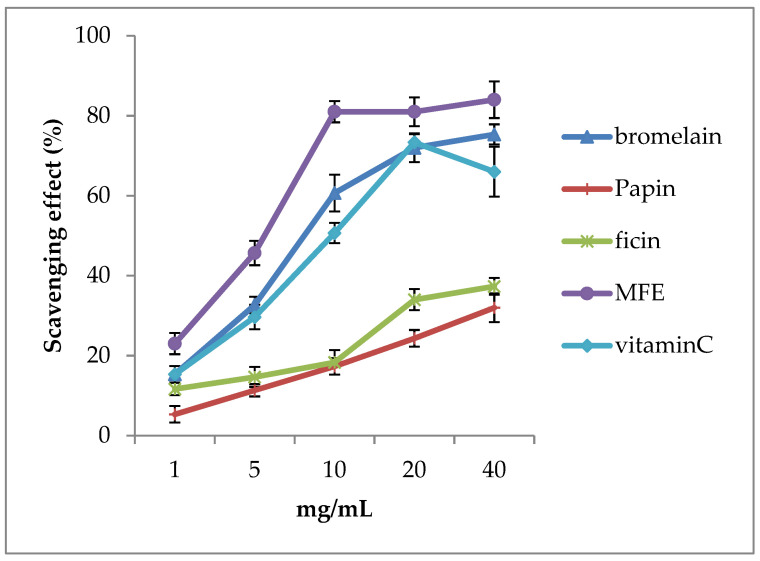
The free radical scavenging activity of various fruit enzymes, including vitamins C (placebo) and mixed fruit enzymes (MFEs). We found that the antioxidant ability of MFEs and bromelain is higher than in vitamin C. It was also revealed that MFEs (purple color line) have significantly higher scavenging activities than vitamin C (sky-blue color line) in all concentrations (1, 5, 10, 20, and 40 μg/mL) (*p* < 0.05). Therefore, we suggested that MFEs enhance higher antioxidant abilities to resist oxidative stress compared with vitamin C. Thus, MFEs could help patients with ocular floaters and intraocular hemorrhage–induced floaters cut and absorb in clinics.

**Figure 4 jcm-11-06710-f004:**
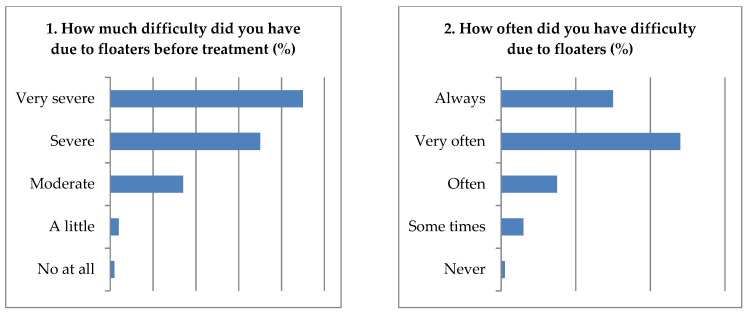
Results from the questionnaire on floaters-induced discomfort; the ocular floaters’ effect on daily activities and satisfaction were assessed.

**Table 1 jcm-11-06710-t001:** The changes in patients with ocular vitreous before and after taking the mixed fruit enzyme capsules over a three-month period in Experiment 1.

**Numbers of Capsules Each Day**	**Before**	**1st Month**	**2nd Month**	**3rd Month**
None	40	40	43	38 (95%)
1 capsule	40	40	36	18 * (45%) #
2 capsules	40	39	34	15 * (37.5) #
3 capsules	40	38	32	12 * (30%) #

Group 1 (N = 40 eyes): no proteases were taken (placebo group) (vitamin C 10 mg daily); Group 2 (N = 40 eyes): all patients took 1 capsule with mixed proteases, including 190 mg bromelain, 95 mg papain, and 95 mg ficin every day; Group 3 (N = 40 eyes): all patients took 2 capsules daily; Group 4 (N = 40 eyes): all patients took 3 capsules daily. *p* < 0.05 was denoted as symbol * by the Scheffe test; *p* < 0.05 was denoted as symbol # by the Williams’ test.

**Table 2 jcm-11-06710-t002:** The changes in ocular floaters before and after taking the mixed fruit enzyme capsules over a three-month period in Experiment 2.

Numbers of Capsules Each Day	Before	1st Month	2nd Month	3rd Month
None	16	15	15	13 (82%)
1 capsule	16	14	13	13 (82%)
2 capsules	16	13	12	12 (75%)
3 capsules	16	13	12	7 * (44%) #

Group 1 (N = 16 eyes): no proteases were taken (placebo group); Group 2 (N = 16 eyes): all patients took 1 capsule with mixed proteases, including 190 mg bromelain, 95 mg papain, and 95 mg ficin daily; Group 3 (N = 16 eyes): all patients took 2 capsules daily; Group 4 (N = 16 eyes): all patients took 3 capsules daily. *p* < 0.05 was denoted as symbol * by the Scheffe test; *p* < 0.05 was denoted as symbol # by the Williams’ test.

**Table 3 jcm-11-06710-t003:** The assessments of the patients’ vision before and after taking the mixed fruit enzyme capsules over a three-month period in Experiment 2.

Numbers of Capsules Each Day	Before	1st Month	2nd Month	3rd Month
None	0.61 ± 0.11	0.62 ± 0.12	0.58 ± 0.10	0.59 ± 0.15
1 capsule	0.59 ± 0.08	0.54 ± 0.15	0.48 ± 0.19	0.40 ± 0.12
2 capsules	0.62 ± 0.15	0.50 ± 0.20	0.45 ± 0.20	0.34 ± 0.15
3 capsules	0.63 ± 0.19	0.49 ± 0.15	0.40 ± 0.25	* 0.19 ± 0.09 #

Vision was presented as log MAR; Log MAR = logarithm of the minimal angle of resolution.; Group 1 (N = 16 eyes): no proteases were taken (placebo group); Group 2 (N = 16 eyes): all patients took 1 capsule with mixed proteases, including 190 mg bromelain, 95 mg papain, and 95 mg ficin daily; Group 3 (N = 16 eyes): all patients took 2 capsules daily; Group 4 (N = 16 eyes): all patients took 3 capsules daily. *p* < 0.05 was denoted as symbol * by the Scheffe test; *p* < 0.05 was denoted as symbol # by the Williams’ test.

## Data Availability

All data generated or analyzed during this study are included in this published article.

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
