# Peer review of "A New Pharmacological Vitreolysis through the Supplement of Mixed Fruit Enzymes for Patients with Ocular Floaters or Vitreous Hemorrhage-Induced Floaters"

_jcm, 2022, doi:10.3390/jcm11226710_

Round 1

Reviewer 1 Report (Previous Reviewer 1)

Line 17.

It reads: “Methods: 224 patients with monocular symptomatic vitreous opacities (SVOs) were recruited between September and December 2017, and received oral supplementation of MFEs (190 mg bromelain, 95 mg papain, and 95 mg ficin) for 3 months double blind clinical trials”.

COMMENT

It should read: “Methods: 224 patients with monocular symptomatic vitreous opacities (SVOs) were recruited between September and December 2017, and received oral supplementation of MFEs (190 mg bromelain, 95 mg papain, and 95 mg ficin) for 3 months in a double-blind clinical trial”.

Line 481. It reads: “This study firstdeveloped the method that employs fruit enzymes to clear and absorb eye floaters”.

COMMENT

It should read: “This study reports the results of a newly developed method that employs fruit enzymes oral supplementation to treat eye floaters”.

Author Response

Dear Reviewer,

Thank you for the precious suggestions on the manuscript, “A New Pharmacological Vitreolysis through the Supplement of Mixed Fruit Enzymes for Patients with Ocular Floaters or Vitreous Hemorrhage–Induced Floaters,” and giving us the opportunity to revise and improve the content of it. We have made substantial revisions to the manuscript to highlight the novelty and potential of our results based on the reviewer’ comments, as the details below.

  1. Line 17. It reads: “Methods: 224 patients with monocular symptomatic vitreous opacities (SVOs) were recruited between September and December 2017, and received oral supplementation of MFEs (190 mg bromelain, 95 mg papain, and 95 mg ficin) for 3 months double blind clinical trials”. COMMENT: It should read: “Methods: 224 patients with monocular symptomatic vitreous opacities (SVOs) were recruited between September and December 2017, and received oral supplementation of MFEs (190 mg bromelain, 95 mg papain, and 95 mg ficin) for 3 months in a double-blind clinical trial”.

 Response: The sentence has been modified according to the comment (lines 17-19).

  1. Line 481. It reads: “This study first developed the method that employs fruit enzymes to clear and absorb eye floaters” .COMMENT: It should read: “This study reports the results of a newly developed method that employs fruit enzymes oral supplementation to treat eye floaters”.

Response: The sentence has been modified according to the comment (lines 608-610).

With above responses and substantial revisions, please kindly consider its publication in J. Clin. Med. In case there is any additional issue that shall be addressed, please feel free to contact me. Thank you for your kind consideration and assistance.

Sincerely,

Chi-Ting Horng

MD, Ph.D.

Reviewer 2 Report (New Reviewer)

The authors investigated the pharmacologic effects of oral supplementation of mixed fruit enzymes for treatment spontaneous symptomatic vitreous opacities and those secondary to vitreous hemorrhage and they found that high dose oral supplementation (bromelain, papain and ficin) was effective in reducing vitreous opacities, even after intraocular hemorrhage. In addition MEF improve patients’ satisfaction and visual acuity.

The introduction is extremely long and give excessive and not essential information about the human vitreous, floaters. Give a general information about the vitreous, floaters and center the introduction in the potential treatment about vitreous floaters.

The method has to be section in design, subjects, materials, procedure and statistically analysis.

The excess of 4 groups implies a reduction of the statistics power

Give extra information about the questionnaire on satisfaction and subjective sensation and its if validated in previous research.

In the results, the tables and graphs should be inserted along the manuscript, not at the end of the manuscript.

The manuscript was well-written, and design and the results are of the interest of the scientific community.

I recommend dividing the discussion into section to improve the comprehension

429 – 460 paragraph should be revised and rewrite to clarify the findings and the discussion points with other authors.

Include a limitation section and a future research of study section that give the potential readers a possible to find a gap in the research to performed future studies.

Rewrite the conclusion to give a strong key message of the topic in the manuscript. It is to long, reduce into two or three sentences.

Update the references to be published after 2010 and also to be included in the indexed journals, avoid non indexed journals.

Author Response

Dear Reviewer,

Thank you for the precious suggestions on the manuscript, “A New Pharmacological Vitreolysis through the Supplement of Mixed Fruit Enzymes for Patients with Ocular Floaters or Vitreous Hemorrhage–Induced Floaters,” and giving us the opportunity to revise and improve the content of it. We have made substantial revisions to the manuscript to highlight the novelty and potential of our results based on the reviewer’ comments, as the details below. 

The authors investigated the pharmacologic effects of oral supplementation of mixed fruit enzymes for treatment spontaneous symptomatic vitreous opacities and those secondary to vitreous hemorrhage and they found that high dose oral supplementation (bromelain, papain and ficin) was effective in reducing vitreous opacities, even after intraocular hemorrhage. In addition MEF improve patients’ satisfaction and visual acuity.

  1. The introduction is extremely long and give excessive and not essential information about the human vitreous, floaters. Give a general information about the vitreous, floaters and center the introduction in the potential treatment about vitreous floaters.

 Response: The introduction has been shorten as much as possible, and presented general information of the vitreous, floaters and the treatments of vitreous floaters.

  1. The method has to be section in design, subjects, materials, procedure and statistically analysis.

Response: The method has been divided into design, subjects, materials, procedure and statistically analysis sections.

  1. The excess of 4 groups implies a reduction of the statistics power.

Response: Thanks for the comment. There were exactly 4 groups in this study, and  anything excess of 4 groups would have reduced the statistics power.

  1. Give extra information about the questionnaire on satisfaction and subjective sensation and its if validated in previous research.

Response: All the information about the questionnaire on satisfaction and subjective sensation were already shown in the Fig. 4 and section 3.3. This study reported the results of a newly developed method that employs fruit enzymes oral supplementation to treat eye floaters, which was not validated in previous research.

  1. In the results, the tables and graphs should be inserted along the manuscript, not at the end of the manuscript.

Response: The tables and graphs have been inserted along the manuscript.

  1. The manuscript was well-written, and design and the results are of the interest of the scientific community. I recommend dividing the discussion into section to improve the comprehension.

Response: Thanks for the comment. The discussion has been rewritten and improved.

  1. 429 – 460 paragraph should be revised and rewrite to clarify the findings and the discussion points with other authors. Include a limitation section and a future research of study section that give the potential readers a possible to find a gap in the research to performed future studies.

Response: Thanks for the comment. Lines 429-460 have been revised and rewritten to clarify the findings and the discussion points with all authors.

Limitation of this study have been added in lines 601-605.

Some limitations were found in this study. First, the mechanisms of MEFs dissolved and absorbed SVO and VH need to be further investigated. Second, SVO for a few patients still persisted after three-months of MEFs treatment, as some types of collagen fibers forming vitreous opacity could not be absorbed and resulted in the disappearance rate of SVO below100%.

  1. Rewrite the conclusion to give a strong key message of the topic in the manuscript. It is to long, reduce into two or three sentences.

Response: The conclusion has been rewritten in lines 608-610.

  1. Update the references to be published after 2010 and also to be included in the indexed journals, avoid non indexed journals.

Response: The references have been updated, and 9 new references added in the manuscript.

With above responses and substantial revisions, please kindly consider its publication in J. Clin. Med. In case there is any additional issue that shall be addressed, please feel free to contact me. Thank you for your kind consideration and assistance.

Sincerely,

Chi-Ting Horng

MD, Ph.D.

Round 2

Reviewer 2 Report (New Reviewer)

The authors solve the comments

This manuscript is a resubmission of an earlier submission. The following is a list of the peer review reports and author responses from that submission.

Round 1

Reviewer 1 Report

A New Pharmacological Vitreolysis through the Supplement of Mixed Fruit Enzymes for Patients with Ocular Floaters or Vitreous Hemorrhage–Induced Floaters.

Line 13. It reads: “Our survey investigated the pharmacologic effects of mixed fruit enzymes (MFEs) for treating vitreous opacities and vitreous hemorrhage (VH) for 3 months.”

Comment: Consider modifying to: “Our study investigated the pharmacologic effects of oral supplementation of mixed fruit enzymes (MFEs) for treating spontaneous symptomatic vitreous opacities (SVOs) and those secondary to vitreous hemorrhage (VH).”

Line 14. It reads: “Methods: The study was performed between September 2017 and December 2017. All 224 subjects with 224 eyes perceiving symptomatic vitreous opacities (SVOs) were recruited. Each capsule contained 190 mg bromelain, 95 mg papain, and 95 mg ficin.”.

Comment: Consider modifying to: “Methods: 224 patients with monocular symptomatic vitreous opacities (SVOs) were recruited between September and December 2017, and received oral supplementation of MFEs (190 mg bromelain, 95 mg papain, and 95 mg ficin) for 3 months.” In addition, it is necessary to clarify that the study was a clinical trial, and describe the blinding plan used (single-blind, double-blind, and triple-blind).

Line 17. It reads: “All volunteers took the doses according to our project in experiments 1 and 2 for 3 months. Different shapes of ocular floaters were considered similar. First, we counted the numbers of SVOs at the beginning of the study, the 1st month, the 2nd month, and the 3 rd month. In experiment 1, 160 participants with SVOs were randomly classified into placebo (10 mg/day vitamin C), Group 1 (1 capsule/day), Group 2 (2 capsules/day), and Group 3 (3 capsules/day). Meanwhile, in experiment 2, 64 eyes with VH-induced SVOs were also divided into four groups, including placebo (10 mg/day vitamin C), Group 1 (1 capsule/day), Group 2 (2 capsules/day), and Group 3 (3 capsules/day). Further, in experiment 2, the changes in the patients’ vision were assessed after 3 months.

Comment: Consider modifying to: “Participants were divided according to the etiology of the SVOs, spontaneous (experiment 1) versus VH (experiment 2), and then randomly assigned into four treatments groups: one group received oral vitamin C, as a placebo, and the other 3 groups received 1 capsule per day (low dose), 2 capsules per day (middle dose), or 3 capsules per day (high dose) of MFEs. The number of SVOs was determined at baseline and then 1, 2, and 3 months after initiating treatment. Further, in cases secondary to VH, the changes in corrected distance visual acuity (CDVA) were assessed after 3 months.”

Line 25. It reads: “Second, we compared the free radical scavenging activities of vitamin C, bromelain, papain, ficin, and MFEs by DDPH assay to investigate their antioxidant abilities”.

Comment: Consider modifying to: “Second, we compared the free radical scavenging capabilities of each substance: vitamin C, bromelain, papain, ficin, and MFEs (combination of bromelain, papain, and ficin), by DDPH assay”.

Line 27. It reads: “Finally, at the end of this study, SVOs-induced patients’ discomfort, the daily activities affected by SVOs, and satisfaction with our treatments were evaluated through 6 questionnaires”.

Comment: Consider modifying to: “Finally, SVOs related symptoms and satisfaction with the treatments were evaluated at the last follow-up visit”.

Line 30. It reads: “capsules daily, respectively (total P<0.05), showing a dose-dependent manner.”

Comment:

The exact p-value should be shown, except if it is too small, in which case the "less than" sign (<) can be used and display a value with three decimal places (for example: <0.001). In all other cases, the exact value of p must be indicated. This change must be made throughout the entire manuscript.

Line 31. It reads: “SVOs was 18% (P> 0.05)”.

Comment: The exact p-value should be shown, except if it is too small, in which case the "less than" sign (<) can be used and display a value with three decimal places (for example: <0.001). In all other cases, the exact value of p must be indicated. This change must be made throughout the entire manuscript.

Line 32. It reads: “It revealed that only taking 3 capsules daily might significantly decrease 56% of VH three months later.”.

Comment:

Since the p values are shown, the sentence “It revealed that only taking 3 capsules daily might significantly decrease 56% of VH three months later” is redundant and should be deleted from the “Abstract”. It can be included in “Discussion” section.

Line 33. It reads: “In addition, the patients’ vision elevated from 0.63LogMAR to 0.19LogMAR (P < 0.05).”

Comment: The exact p-value should be shown, except if it is too small, in which case the "less than" sign (<) can be used and display a value with three decimal places (for example: <0.001). In all other cases, the exact value of p must be indicated. This change must be made throughout the entire manuscript. CDVA is the term to be used. Therefore, it should read: “In experiment 2,  CDVA improved from 0.63 LogMAR to 0.19 LogMAR (P = 0.0xxxx).”

Line 34. It reads: “This method undertaken by our study is the first in the world that clears eye floaters using fruit enzymes. We suggested that the mixtures of bromelain, papain, and ficin might decrease ocular floaters and even intraocular hemorrhage.”.

Comment:

Authors' comments about the approach should be reserved for “Discussion” section. In addition, medical terms should always be preferred to non-medical terms.

Therefore, consider modifying to: “A pharmacological approach using high dose of oral supplementation with MFEs (bromelain, papain, and ficin) was effective in reducing vitreous opacities, even after intraocular hemorrhage.”

Line 36. It reads: “Furthermore, MEFs should enhance the disappearance of cellular debris and blood clots in the vitreous cavity, improving patients’ vision and satisfaction with pharmacologic vitreolysis”.

Comment:

The action mechanism of the pharmacological approach is yet to be defined. Therefore, consider modifying to: “Furthermore, pharmacologic vitreolysis with MEFs supplementation showed high patients´ satisfaction, and also improved CDVA in patients with vitreous hemorrhage–induced floaters”.  

Line 46. It reads: “During aging, pathologic processes and oxidative stress occur, leading to the depolymerization of HA, loss of water, liquefaction of vitreous, and disordered collagen fibrils”.

Comment:

It should read: “With aging, pathological processes and oxidative stress occur, leading to HA depolymerization, water loss, vitreous liquefaction, and disorganization of collagen fibrils.”

Line 51. It reads: “At present, opacities are projected to the retina and interpreted as a “flying mosquito” in the brain”.

Comment:

It should read: “These opacities in the vitreous are projected onto the retina and are interpreted by the brain as moving objects ("eye floaters" or "muscae volitantes" which in Latin means "flying flies").

Line 53. It reads: “In addition, collagen fibers I, III, IV, and XVIII have been described as components of retinal vasculature.”

Comment:

This sentence is not related to the specific topic and should be deleted.

Line 62. It reads: “minimalist decors are their major sources.”.

Comment: It should read: “The minimalist decorations, with monochrome backgrounds and with few objects, frequently put them in evidence”.

Line 63. It reads. “Initially, this symptom appears annoying, but the brain eventually adapts and forgets its existence for at least 1 week to 5 years”.

Comment:

It should read: “Initially, the symptoms can be very bothersome, but the brain eventually adapts to them and the patient becomes less aware of their existence, but this process can take from 1 week to several years, and in some cases the intense symptoms do not diminish. never".

Line 67. It reads: “Bond-Taylor et al. found that when subjects have an acute onset of floaters, the prevalence of retinal tear or retinal break is around 14%, VH is 22.7%, and retinal detachment (RD) is up to 13.5% [6].”

Comment:

Consider modifying to: “The symptoms are not the only problem of vitreous condensation, but its coexistence with other secondary alterations in the retina. Bond-Taylor et al. found that when subjects had an acute onset of floaters, the prevalence of retinal tears or retinal breaks was around 14%, HV was 22.7%, and retinal detachment (RD) prevalence was up to 13%.5% [6]”.

Line 69. It reads: “Chronic floaters were also found to have a similar degree of misery index as cataracts and macular membranes, affecting visual function [7]”.

Comment:  This sentence is confusing, and the reference cited has to do with the association of floaters with other retinal conditions, which has already been discussed. The phrase should be removed.

Line 73. It reads: “In the past, ocular floaters would occur among individuals aged 40— _most of them experiencing floaters in their remaining life span. Now, the trend of victims is becoming younger because of the blue light widely emitted from electronic screens, such as iPads, smartphones, and liquid crystal displays (LCDs). Chen et al. suggested that the energy from blue light might accelerate vitreous degeneration, resulting in vitreous opacity and increased ocular floaters, affecting 39.7% of users[9].”

Comment:

Consider modifying to: “In the past, ocular floaters usually occurred among people over 40 years of age, but it seems that they are now occurring in younger people. It has been hypothesized that this is related to the blue light that is widely emitted by electronic displays, such as iPads, smartphones, and liquid crystal displays (LCDs). Chen et al. suggested that blue light energy could accelerate vitreous degeneration, resulting in increased vitreous opacity and increased ocular floaters, potentially affecting up to 39.7% of users[9 ]”.

Lines 85 to 241.

Comment:  “Introduction” section is by far too long. All these paragraphs should be summarized in only about three paragraphs. Excessive details should be eliminated, and only general ideas should be left, otherwise, the reader will be overwhelmed because the information is excessive. It is essential that a person whose native language is English review these new paragraphs.

Line 242. It reads: “Recently, we developed a new treatment method involving pharmacologic vitreolysis by combining bromelain, papain, and ficin that cures patients with SVOs [45]. In this study, we wanted to furtherly focus on the effectiveness of MFE capsules for patients with various SVOs and VH-induced SVOs. Furthermore, we assessed the antioxidant ability of individual fruit enzymes and MFEs in the laboratory. Moreover, we attempted to inquire through 6 questionnaires about patients’ impression of oular floaters, influence level of daily activities, and the satisfaction of the new method that employs mixed fruit enzyme vitreolysis”.

Comment.

Reference cited (45. Cordero, I. Understanding and safety using ophthalmic lasers. Community. Eye. Health. 2015,28,76-77) does not match the text. It seems that it should correspond to references # 50 or  #52.

The word “cure” does not really fit in the text.

 Consider modifying to “Recently, we developed a new treatment method involving pharmacologic vitreolysis by combining bromelain, papain, and ficin, with which we successfully treated patients with SVOs. [REFERENCES]. In this study, we wanted to furtherly focus on the effectiveness of MFE capsules for patients with both primary and VH-induced SVOs. Furthermore, we evaluated the antioxidant ability of individual fruit enzymes and MFEs in the laboratory. Moreover, we assessed with a questionnaire patients’ impression of ocular floaters, influence level on daily activities, and the satisfaction of the new therapeutic method with MFEs”.

Line 250. In “Methods and Materials”.

Comment: At the beginning of the "Methods and Materials" section, it must be made clear that it was a clinical trial, and it must be indicated the blinding plan used (single-blind, double-blind, and triple-blind).

Line 260. It reads: “The age, gender, and the number of ocular floaters were recorded from each patient simultaneously.”

Comment: This sentence is redundant and should be deleted.

Line 272. It reads: “In other words, only volunteers with VH due to DR, hypertensive retinopathy, and ocular trauma were recruited.”

Comment: Does this mean that patients with HV secondary to spontaneous posterior vitreous detachment (PVD) were also excluded? It seems to be a typing error.

Line 274. It reads: “The chemical substances in our method employing pharmacologic vitreolysis are derived from three natural fruit enzymes. The bromelain powder was obtained from Xi’an Sgneck Biologic Technology Co., Ltd. (China), the papain powder was purchased from Shaani Hongda Phytochemistry Co., Ltd. (China), and the ficin extract was from HK. Gotopharm Co., Ltd. (Taiwan)”.

Comment: The information on the origin of the substances is incomplete, and the reader is left with gaps. Consider modifying to:

“The chemical substances in our method employing pharmacologic vitreolysis are derived from three natural enzymes: bromelain from pineapple, papain from papaya, and ficin obtained from latex from the trunk of a tropical fig tree. The suppliers of the purified substances were: for bromelain (powder), Xi'an Sgneck Biologic Technology Co., Ltd. (China); for papain (powder), Shaani Hongda Phytochemistry Co., Ltd. (China), and  for ficin (extract), HK. Gotopharm Co., Ltd. (Taiwan)”.

Line 278. It reads. “According to the gold ratio from our previous study, each capsule contains 190mg bromelain, 95mg papain, and 95mg ficin [47].”

Comment: The cited reference does not match the text:

47. Roth, M.; Trittibach, P.; Koerner, F.; Sarra, G. Pars plana vitrectomy for idiopathic vitreous floater. Klin. Monatsbl. Augenheilkd. 847 2005, 222, 728-732

In addition, consider modifying to : “According to our previous study, each capsule was prepared to contain 190 mg bromelain, 95 mg papain, and 95 mg ficin [REFERENCE].”

Line 289. It reads: “(one group consisted of 16 subjects). In addition, the blood sugar and blood pressure of the patients with DM or hypertension is well controlled by physicians throughout the entire study.”.

Comment: It should read: “(each group consisted of 16 subjects). In addition, the blood sugar and blood pressure of the patients with DM or hypertension was controlled well by medical treatment throughout the study.”

Line 296. It reads: “Furthermore, we also compared the change of best-corrected visual acuity (BCVA) after 3-month fruit enzyme intake in experiment 2.”

Comment:

It should read: “Furthermore, we also compared the change in corrected distance visual acuity (CDVA) after 3-month MFEs intake in experiment 2.”

Reference for the change in visual acuity terms: Dupps WJ Jr, Kohnen T, Mamalis N, et al. Standardized graphs and terms for refractive surgery results. J Cataract Refract Surg. 2011;37(1):1-3. doi:10.1016/j.jcrs.2010.11.010

Line 299. It reads: “In clinics, many individuals had perceived a “floating mosquito”. However, eye doctors at present could not observe anything in the vitreous cavity. Hence, the numbers of  SVOs could not be qualified during experiments 1 and 2. To avoid bias, we selected the participants with subjective floaters and combined them with the objective findings from ocular machines that were consistent simultaneously.”.

Comment:

The sentences are confusing.

Consider modifying to the following sentence (if that's really what the authors mean): “In real life, many individuals perceive a  vitreous floater, but ophthalmologist does not find any abnormality in the vitreous cavity. Hence, the numbers of SVOs could not be easily objectively identified. To avoid bias, during experiments 1 and 2, we selected the participants with subjective floaters that were  simultaneously detected with ocular diagnostic devices.”

Line 305. It reads: “The vitreous opacity was evaluated by non-mydriatic retinal photography (Kowa, E-vision Co., Ltd.), Ultra Scan-B (Alcon Co., Ltd.), and optic coherence tomography (OCT) (Clinico Co., Ltd.).

Comment:

It should read:” The vitreous opacity was evaluated by non-mydriatic retinal photography (Kowa, E-vision Co., Ltd.), B-mode ultrasonography (Ultra Scan-B, Alcon Co., Ltd.), and optic coherence tomography (OCT, Clinico Co., Ltd.).”

Line 308. It reads: “any one image with a different shape was calculated and considered the same”

Comment.

This sentence is not clear.

I stopped the review here because the article is actually too long (by far). The subject matter is interesting, but there are many writing problems in English.

The "Introduction" and "Discussion" should be reduced to approximately one-third of their current length.

One possibility for authors to explore is to separate the clinical study from the laboratory study so that it is a more manageable study for the reader.

Author Response

Dear Reviewer,

Thank you for the precious suggestions on the manuscript, “A New Pharmacological Vitreolysis through the Supplement of Mixed Fruit Enzymes for Patients with Ocular Floaters or Vitreous Hemorrhage–Induced Floaters,” and giving us the opportunity to revise and improve the content of it. We have made substantial revisions to the manuscript to highlight the novelty and potential of our results based on the reviewer’ comments, as the details below.

  1. Line 13. It reads: “Our survey investigated the pharmacologic effects of mixed fruit enzymes (MFEs) for treating vitreous opacities and vitreous hemorrhage (VH) for 3 months. ”Comment: Consider modifying to: “Our study investigated the pharmacologic effects of oral supplementation of mixed fruit enzymes (MFEs) for treating spontaneous symptomatic vitreous opacities (SVOs) and those secondary to vitreous hemorrhage (VH).”

Response: The sentence has been modified according to the comment (lines 13-14).

  1. Line 14. It reads: “Methods: The study was performed between September 2017 and December 2017. All 224 subjects with 224 eyes perceiving symptomatic vitreous opacities (SVOs) were recruited. Each capsule contained 190 mg bromelain, 95 mg papain, and 95 mg ficin.”. Comment: Consider modifying to: “Methods: 224 patients with monocular symptomatic vitreous opacities (SVOs) were recruited between September and December 2017, and received oral supplementation of MFEs (190 mg bromelain, 95 mg papain, and 95 mg ficin) for 3 months.” In addition, it is necessary to clarify that the study was a clinical trial, and describe the blinding plan used (single-blind, double-blind, and triple-blind).

 Response: The sentences have been modified according to the comment (lines 17-19). This study has been described and clarified as a double blind clinical trials.

  1. Line 17. It reads: “All volunteers took the doses according to our project in experiments 1 and 2 for 3 months. Different shapes of ocular floaters were considered similar. First, we counted the numbers of SVOs at the beginning of the study, the 1st month, the 2nd month, and the 3 rd month. In experiment 1, 160 participants with SVOs were randomly classified into placebo (10 mg/day vitamin C), Group 1 (1 capsule/day), Group 2 (2 capsules/day), and Group 3 (3 capsules/day). Meanwhile, in experiment 2, 64 eyes with VH-induced SVOs were also divided into four groups, including placebo (10 mg/day vitamin C), Group 1 (1 capsule/day), Group 2 (2 capsules/day), and Group 3 (3 capsules/day). Further, in experiment 2, the changes in the patients’ vision were assessed after 3 months. Comment: Consider modifying to: “Participants were divided according to the etiology of the SVOs, spontaneous (experiment 1) versus VH (experiment 2), and then randomly assigned into four treatments groups: one group received oral vitamin C, as a placebo, and the other 3 groups received 1 capsule per day (low dose), 2 capsules per day (middle dose), or 3 capsules per day (high dose) of MFEs. The number of SVOs was determined at baseline and then 1, 2, and 3 months after initiating treatment. Further, in cases secondary to VH, the changes in corrected distance visual acuity (CDVA) were assessed after 3 months.”

 Response: The sentences have been modified according to the comment (lines 22-27). 

  1. Line 25. It reads: “Second, we compared the free radical scavenging activities of vitamin C, bromelain, papain, ficin, and MFEs by DDPH assay to investigate their antioxidant abilities”. Comment: Consider modifying to: “Second, we compared the free radical scavenging capabilities of each substance: vitamin C, bromelain, papain, ficin, and MFEs (combination of bromelain, papain, and ficin), by DDPH assay”.

 Response: The sentences have been modified according to the comment (lines 39-40). 

  1. Line 27. It reads: “Finally, at the end of this study, SVOs-induced patients’ discomfort, the daily activities affected by SVOs, and satisfaction with our treatments were evaluated through 6 questionnaires”. Comment: Consider modifying to: “Finally, SVOs related symptoms and satisfaction with the treatments were evaluated at the last follow-up visit”.

  Response: The sentence has been modified according to the comment (lines 22-27). 

  1. Line 30. It reads: “capsules daily, respectively (total P<0.05), showing a dose-dependent manner.”

Comment: The exact p-value should be shown, except if it is too small, in which case the "less than" sign (<) can be used and display a value with three decimal places (for example: <0.001). In all other cases, the exact value of p must be indicated. This change must be made throughout the entire manuscript.

  Response: All the exact values of p have be indicated in the manuscript. 

  1. Line 31. It reads: “SVOs was 18% (P> 0.05)”.Comment: The exact p-value should be shown, except if it is too small, in which case the "less than" sign (<) can be used and display a value with three decimal places (for example: <0.001). In all other cases, the exact value of p must be indicated. This change must be made throughout the entire manuscript.

 Response: All the exact values of p have be indicated in the manuscript. 

  1. Line 32. It reads: “It revealed that only taking 3 capsules daily might significantly decrease 56% of VH three months later.”.Comment: Since the p values are shown, the sentence “It revealed that only taking 3 capsules daily might significantly decrease 56% of VH three months later” is redundant and should be deleted from the “Abstract”. It can be included in “Discussion” section.

  Response: The sentence “It revealed that only taking 3 capsules daily might significantly decrease 56% of VH three months later” have been deleted from the abstract and included in discussion section.

  1. Line 33. It reads: “In addition, the patients’ vision elevated from 0.63LogMAR to 0.19LogMAR (P < 0.05).”Comment: The exact p-value should be shown, except if it is too small, in which case the "less than" sign (<) can be used and display a value with three decimal places (for example: <0.001). In all other cases, the exact value of p must be indicated. This change must be made throughout the entire manuscript. CDVA is the term to be used. Therefore, it should read: “In experiment 2, CDVA improved from 0.63 LogMAR to 0.19 LogMAR (P = 0.0xxxx).”

  Response: All the exact values of p have be indicated in the manuscript. 

  1. Line 34. It reads: “This method undertaken by our study is the first in the world that clears eye floaters using fruit enzymes. We suggested that the mixtures of bromelain, papain, and ficin might decrease ocular floaters and even intraocular hemorrhage.”.

Comment:Authors' comments about the approach should be reserved for “Discussion” section. In addition, medical terms should always be preferred to non-medical terms.

Therefore, consider modifying to: “A pharmacological approach using high dose of oral supplementation with MFEs (bromelain, papain, and ficin) was effective in reducing vitreous opacities, even after intraocular hemorrhage.”

   Response: The sentences have been modified according to the comment (lines 48-50). 

  1. Line 36. It reads: “Furthermore, MEFs should enhance the disappearance of cellular debris and blood clots in the vitreous cavity, improving patients’ vision and satisfaction with pharmacologic vitreolysis”.Comment: The action mechanism of the pharmacological approach is yet to be defined. Therefore, consider modifying to: “Furthermore, pharmacologic vitreolysis with MEFs supplementation showed high patients´ satisfaction, and also improved CDVA in patients with vitreous hemorrhage–induced floaters”.  

  Response: The sentences have been modified according to the comment (lines 52-54).  

  1. Line 46. It reads: “During aging, pathologic processes and oxidative stress occur, leading to the depolymerization of HA, loss of water, liquefaction of vitreous, and disordered collagen fibrils”.

Comment: It should read: “With aging, pathological processes and oxidative stress occur, leading to HA depolymerization, water loss, vitreous liquefaction, and disorganization of collagen fibrils.”

  Response: The sentences have been modified according to the comment (lines 65-67).

  1. Line 51. It reads: “At present, opacities are projected to the retina and interpreted as a “flying mosquito” in the brain”.Comment: It should read: “These opacities in the vitreous are projected onto the retina and are interpreted by the brain as moving objects ("eye floaters" or "muscae volitantes" which in Latin means "flying flies").

 Response: The sentences have been modified according to the comment (lines 71-72).

  1. Line 53. It reads: “In addition, collagen fibers I, III, IV, and XVIII have been described as components of retinal vasculature. ”Comment: This sentence is not related to the specific topic and should be deleted.

 Response: The sentence has been deleted according to the comment.

  1. Line 62. It reads: “minimalist decors are their major sources. ”Comment: It should read: “The minimalist decorations, with monochrome backgrounds and with few objects, frequently put them in evidence”.

  Response: The sentence has been modified according to the comment (lines 83-84).

  1. Line 63. It reads. “Initially, this symptom appears annoying, but the brain eventually adapts and forgets its existence for at least 1 week to 5 years”. Comment: It should read: “Initially, the symptoms can be very bothersome, but the brain eventually adapts to them and the patient becomes less aware of their existence, but this process can take from 1 week to several years, and in some cases the intense symptoms do not diminish. never".

Response: The sentence has been modified according to the comment (lines 85-88).

  1. Line 67. It reads: “Bond-Taylor et al. found that when subjects have an acute onset of floaters, the prevalence of retinal tear or retinal break is around 14%, VH is 22.7%, and retinal detachment (RD) is up to 13.5% [6].”Comment: Consider modifying to: “The symptoms are not the only problem of vitreous condensation, but its coexistence with other secondary alterations in the retina. Bond-Taylor et al. found that when subjects had an acute onset of floaters, the prevalence of retinal tears or retinal breaks was around 14%, HV was 22.7%, and retinal detachment (RD) prevalence was up to 13%.5% [6]”.

 Response: The sentence has been modified according to the comment (lines 92-96).

  1. Line 69. It reads: “Chronic floaters were also found to have a similar degree of misery index as cataracts and macular membranes, affecting visual function [7]”.

Comment:  This sentence is confusing, and the reference cited has to do with the association of floaters with other retinal conditions, which has already been discussed. The phrase should be removed.

 Response: The sentence has been removed according to the comment (lines 92-96).

  1. Line 73. It reads: “In the past, ocular floaters would occur among individuals aged 40— _most of them experiencing floaters in their remaining life span. Now, the trend of victims is becoming younger because of the blue light widely emitted from electronic screens, such as iPads, smartphones, and liquid crystal displays (LCDs). Chen et al. suggested that the energy from blue light might accelerate vitreous degeneration, resulting in vitreous opacity and increased ocular floaters, affecting 39.7% of users[9].”Comment: Consider modifying to: “In the past, ocular floaters usually occurred among people over 40 years of age, but it seems that they are now occurring in younger people. It has been hypothesized that this is related to the blue light that is widely emitted by electronic displays, such as iPads, smartphones, and liquid crystal displays (LCDs). Chen et al. suggested that blue light energy could accelerate vitreous degeneration, resulting in increased vitreous opacity and increased ocular floaters, potentially affecting up to 39.7% of users [9 ]”.

  Response: The sentence has been modified according to the comment (lines 102-108).

  1. Lines 85 to 241.

Comment:  “Introduction” section is by far too long. All these paragraphs should be summarized in only about three paragraphs. Excessive details should be eliminated, and only general ideas should be left, otherwise, the reader will be overwhelmed because the information is excessive. It is essential that a person whose native language is English review these new paragraphs.

  Response: Introduction section have been modified and shorten according to the comment.

  1. Line 242. It reads: “Recently, we developed a new treatment method involving pharmacologic vitreolysis by combining bromelain, papain, and ficin that cures patients with SVOs [45]. In this study, we wanted to furtherly focus on the effectiveness of MFE capsules for patients with various SVOs and VH-induced SVOs. Furthermore, we assessed the antioxidant ability of individual fruit enzymes and MFEs in the laboratory. Moreover, we attempted to inquire through 6 questionnaires about patients’ impression of oular floaters, influence level of daily activities, and the satisfaction of the new method that employs mixed fruit enzyme vitreolysis”.Comment. Reference cited (45. Cordero, I. Understanding and safety using ophthalmic lasers. Community. Eye. Health. 2015,28,76-77) does not match the text. It seems that it should correspond to references # 50 or  #52.

The word “cure” does not really fit in the text. Consider modifying to “Recently, we developed a new treatment method involving pharmacologic vitreolysis by combining bromelain, papain, and ficin, with which we successfully treated patients with SVOs. [50, 52]. In this study, we wanted to furtherly focus on the effectiveness of MFE capsules for patients with both primary and VH-induced SVOs. Furthermore, we evaluated the antioxidant ability of individual fruit enzymes and MFEs in the laboratory. Moreover, we assessed with a questionnaire patients’ impression of ocular floaters, influence level on daily activities, and the satisfaction of the new therapeutic method with MFEs”.

 Response: The sentence has been modified according to the comment (lines 278-284).

  1. Line 250. In “Methods and Materials”.

Comment: At the beginning of the "Methods and Materials" section, it must be made clear that it was a clinical trial, and it must be indicated the blinding plan used (single-blind, double-blind, and triple-blind).

Response: The experiments were double blind clinical trials andperformed between September 2017 and December 2017(294-295).

  1. Line 260. It reads: “The age, gender, and the number of ocular floaters were recorded from each patient simultaneously. ”Comment: This sentence is redundant and should be deleted.

  Response: The sentence has been deleted.

  1. Line 272. It reads: “In other words, only volunteers with VH due to DR, hypertensive retinopathy, and ocular trauma were recruited.”

Comment: Does this mean that patients with HV secondary to spontaneous posterior vitreous detachment (PVD) were also excluded? It seems to be a typing error.

  Response: The sentence has been deleted.

  1. Line 274. It reads: “The chemical substances in our method employing pharmacologic vitreolysis are derived from three natural fruit enzymes. The bromelain powder was obtained from Xi’an Sgneck Biologic Technology Co., Ltd. (China), the papain powder was purchased from Shaani Hongda Phytochemistry Co., Ltd. (China), and the ficin extract was from HK. Gotopharm Co., Ltd. (Taiwan)”.Comment: The information on the origin of the substances is incomplete, and the reader is left with gaps. Consider modifying to:“The chemical substances in our method employing pharmacologic vitreolysis are derived from three natural enzymes: bromelain from pineapple, papain from papaya, and ficin obtained from latex from the trunk of a tropical fig tree. The suppliers of the purified substances were: for bromelain (powder), Xi'an Sgneck Biologic Technology Co., Ltd. (China); for papain (powder), Shaani Hongda Phytochemistry Co., Ltd. (China), and  for ficin (extract), HK. Gotopharm Co., Ltd. (Taiwan)”.

 Response: The sentences have been modified according to the comment (lines 316-321).

  1. Line 278. It reads. “According to the gold ratio from our previous study, each capsule contains 190mg bromelain, 95mg papain, and 95mg ficin [47].”

Comment: The cited reference does not match the text:

  1. Roth, M.; Trittibach, P.; Koerner, F.; Sarra, G. Pars plana vitrectomy for idiopathic vitreous floater. Klin. Monatsbl. Augenheilkd. 847 2005, 222, 728-732. In addition, consider modifying to : “According to our previous study, each capsule was prepared to contain 190 mg bromelain, 95 mg papain, and 95 mg ficin [REFERENCE].”

  Response: The sentences have been modified according to the comment, and the reference number has been corrected (lines 325-327).

  1. Line 289. It reads: “(one group consisted of 16 subjects). In addition, the blood sugar and blood pressure of the patients with DM or hypertension is well controlled by physicians throughout the entire study.”.Comment: It should read: “(each group consisted of 16 subjects). In addition, the blood sugar and blood pressure of the patients with DM or hypertension was controlled well by medical treatment throughout the study.”

 Response: The sentences have been modified according to the comment (338-340).

  1. Line 296. It reads: “Furthermore, we also compared the change of best-corrected visual acuity (BCVA) after 3-month fruit enzyme intake in experiment 2.”Comment:

It should read: “Furthermore, we also compared the change in corrected distance visual acuity (CDVA) after 3-month MFEs intake in experiment 2.”Reference for the change in visual acuity terms: Dupps WJ Jr, Kohnen T, Mamalis N, et al. Standardized graphs and terms for refractive surgery results. J Cataract Refract Surg. 2011;37(1):1-3. doi:10.1016/j.jcrs.2010.11.010

 Response: The sentences have been modified according to the comment (347-349), and the reference [31] has also been added.

  1. Line 299. It reads: “In clinics, many individuals had perceived a “floating mosquito”. However, eye doctors at present could not observe anything in the vitreous cavity. Hence, the numbers of  SVOs could not be qualified during experiments 1 and 2. To avoid bias, we selected the participants with subjective floaters and combined them with the objective findings from ocular machines that were consistent simultaneously.”.

Comment:The sentences are confusing. Consider modifying to the following sentence (if that's really what the authors mean): “In real life, many individuals perceive a  vitreous floater, but ophthalmologist does not find any abnormality in the vitreous cavity. Hence, the numbers of SVOs could not be easily objectively identified. To avoid bias, during experiments 1 and 2, we selected the participants with subjective floaters that were simultaneously detected with ocular diagnostic devices.”

  Response: The sentences have been modified according to the comment (351-354)

  1. Line 305. It reads: “The vitreous opacity was evaluated by non-mydriatic retinal photography (Kowa, E-vision Co., Ltd.), Ultra Scan-B (Alcon Co., Ltd.), and optic coherence tomography (OCT) (Clinico Co., Ltd.). Comment: It should read:” The vitreous opacity was evaluated by non-mydriatic retinal photography (Kowa, E-vision Co., Ltd.), B-mode ultrasonography (Ultra Scan-B, Alcon Co., Ltd.), and optic coherence tomography (OCT, Clinico Co., Ltd.).”

  Response: The sentences have been modified according to the comment (361-363)

  1. Line 308. It reads: “any one image with a different shape was calculated and considered the same” Comment: This sentence is not clear.

 Response: The sentence has been deleted.

  1. I stopped the review here because the article is actually too long (by far). The subject matter is interesting, but there are many writing problems in English. The "Introduction" and "Discussion" should be reduced to approximately one-third of their current length.

One possibility for authors to explore is to separate the clinical study from the laboratory study so that it is a more manageable study for the reader.

 Response: Thank you for the precious comments again. The manuscript has been edited and shorten.

Reviewer 2 Report

The introduction is too long, as well the methods are not so clear

Discussion has a part that should be inserted in the introduction 

Author Response

Dear Reviewer,

Thank you for the precious suggestions on the manuscript, “A New Pharmacological Vitreolysis through the Supplement of Mixed Fruit Enzymes for Patients with Ocular Floaters or Vitreous Hemorrhage–Induced Floaters,” and giving us the opportunity to revise and improve the content of it. We have made substantial revisions to the manuscript to highlight the novelty and potential of our results based on the reviewer’ comments, as the details below.

  1. The introduction is too long, as well the methods are not so clear.

 Response: The introduction has been shorten, and the methods has been corrected.

  1. Discussion has a part that should be inserted in the introduction 

Response: A part of discussion has been inserted in the introduction, and the manuscript has also been modified and shorten.  

Round 2

Reviewer 2 Report

Poor scientific quality